# Edge Device Detection of Tea Leaves with One Bud and Two Leaves Based on ShuffleNetv2-YOLOv5-Lite-E

**Shihao Zhang [1,2], Hekai Yang [1,2], Chunhua Yang [2,3], Wenxia Yuan [3], Xinghui Li [4], Xinghua Wang [3], Yinsong Zhang [5], Xiaobo Cai [2], Yubo Sheng [6], Xiujuan Deng [3], Wei Huang [3], Lei Li [3], Junjie He [3] and Baijuan Wang [2,3,*]**

1 College of Mechanical and Electrical Engineering, Yunnan Agricultural University, Kunming 650201, China
2 Yunnan Organic Tea Industry Intelligent Engineering Research Center, Yunnan Agricultural University, Kunming 650201, China
3 College of Tea Science, Yunnan Agricultural University, Kunming 650201, China
4 International Institute of Tea Industry Innovation for "The Belt and Road", Nanjing Agricultural University, Nanjing 210095, China
5 College of Foreign Languages, Yunnan Agricultural University, Kunming 650201, China
6 China Tea (Yunnan) Co., Ltd., Kunming 650201, China
* Correspondence: wangbaijuan123@126.com

**Abstract:** In order to solve the problem of an accurate recognition of tea picking through tea picking robots, an edge device detection method is proposed in this paper based on ShuffleNetv2-YOLOv5-Lite-E for tea with one bud and two leaves. This replaces the original feature extraction network by removing the Focus layer and using the ShuffleNetv2 algorithm, followed by a channel pruning of YOLOv5 at the neck layer head, thus achieving the purpose of reducing the model size. The results show that the size of the improved generated weight file is 27% of that of the original YOLOv5 model, and the mAP value of ShuffleNetv2-YOLOv5-Lite-E is 97.43% and 94.52% on the pc and edge device respectively, which are 1.32% and 1.75% lower compared to that of the original YOLOv5 model. The detection speeds of ShuffleNetv2-YOLOv5-Lite-E, YOLOv5, YOLOv4, and YOLOv3 were 8.6 fps, 2.7 fps, 3.2 fps, and 3.4 fps respectively after importing the models into an edge device, and the improved YOLOv5 detection speed was 3.2 times faster than that of the original YOLOv5 model. Through the detection method, the size of the original YOLOv5 model is effectively reduced while essentially ensuring recognition accuracy. The detection speed is also significantly improved, which is conducive to the realization of intelligent and accurate picking for future tea gardens, laying a solid foundation for the realization of tea picking robots.

**Keywords:** ShuffleNetv2-YOLOv5-Lite-E; model size; one bud and two leaves; edge device





## 1. Introduction

The tea in Yunnan is famous for the quality endowed from the unique soil, landform and physiognomy, and climate. With the booming development of the Yunnan tea industry in recent years, the increasing scarcity of labor resources has gradually become the biggest obstacle to the development of the Yunnan tea industry. However, due to the complex and variable terrains of the Yunnan tea plantations, it is extremely difficult to pick tea leaves with large machines. Therefore, the development of tea picking robots and tea picking automation is urgently required by the Yunnan tea industry [1].

After extensive practical investigation by the team, it was found that most of the tea plantations are currently picked by tea farmers manually, which is not only inefficient and takes up much farming time, but omissions also easily occur in the picking process. In the research on Laobanzhang base in Xishuangbanna, Yunnan, it was also found that the resulting waste of tea resources is as high as 15% or more. Therefore, intelligent and precise tea picking has become the most urgent demand in tea production. Under the dual pressure of the insufficient supply of tea pickers and the significant increase of picking costs, the

promotion of intelligent and precise picking in tea plantations is undoubtedly an important means to solving the problem of tea picking.

In order to solve the problems such as indiscriminate mechanical picking, the serious omission of manual picking, and high labor costs, tea picking robots can avoid tea picking omissions as much as is possible while ensuring the accuracy of manual picking. Based on the fact that the network coverage of Yunnan tea gardens is extremely low, the research on transplanting deep learning algorithms based on ShuffleNet v2-YOLOv5-Lite-E to the edge device Raspberry Pie is put forward in this study, so as to accurately identify tea with one bud and two leaves in the natural environment. Through this lightweight YOLOv5 neural network, tea with one bud and two leaves can be accurately identified in real time based on edge equipment without networks [2], which can lay a solid foundation for the intelligent and accurate robot picking of tea gardens in the future. The target recognition methods currently used by commercially-available picking robots are based on features such as color, geometry, and texture, which are to classify, detect, and segment targets. The purpose of these target recognition methods based on color features is to separate the targets from the background using a suitable color model and differentiate pixel color features between the target and background areas. However, with detections using traditional digital image processing methods, high-accuracy and real-time requirements cannot be achieved, and the robustness of traditional methods with artificially-designed features used to encode visual attributes against diverse morphological changes of the targets is insufficient to meet the precise and real-time picking requirements of robots.

Recently, the development of neural networks has been very rapid, and the recognition of shoots has long been integrated into neural network systems. Chen Miaoting et al. trained a model for tea shoots based on a YOLO neural network, whose final recognition accuracy was as high as 84% [3]; Xu Gaojian et al. achieved an accuracy of 85.14% for a tea shoot recognition model after training the Faster R-cnn neural network based on a VGG-16 feature extraction network [4]; and Wang Ziyu et al. achieved an accuracy of 91.5% for tea shoot recognition using an SSD algorithm [5]. The highest recognition rate of tea shoots was achieved through the SSD algorithm of Zijiyu et al. However, the size and computation of the model generated based on the SSD algorithm were relatively large, which was not suitable to be ported to a small mobile device such as a Raspberry Pi, so qualitative changes were not brought to the development of intelligent tea picking robots in that study. Comparing with the improvement of YOLOv5 by experts such as Lele Wang and Yaping Li [6,7], although ShuffleNet was used to lighten the network in their experiment, channel pruning was also used to further process the model in this study. After the lightweight model was generated, it was deployed to a raspberry pie with a camera and was tested in a real tea garden using a tracked robot, where its real detection data were obtained. The mAP values of ShuffleNetv2-YOLOv5-Lite-E are 97.43% and 94.52% on pc and edge device, respectively, which is higher than the final Map value of the improved Yolov5 model in the first two studies.

In most previous studies, the ratios of training set, verification set, and test set are all 8:1:1, and the total dataset is between 500 and 10,000. The experimental results show that the size of the total dataset plays no decisive role in the recognition rate and size of the final model. In order to more accurately test the performance of the model constructed in this experiment in practical application, the ratio of training set to verification set was adjusted to 8:2, and the real-time images taken in the tea garden detection experiment were used as the test set.

Although new YOLO versions have been developed internationally, such as YOLOv7 and YOLOv8, compared with YOLOv5, the updated version of YOLO-series algorithms is rarely used in practice. For example, YOLOv8 presents a new framework, but it is still in its early stage and needs to be improved continuously. This study is an improvement based on YOLOv5 version 6.0, which is more stable and has been highly evaluated in practical application.

In order to ensure that no significant degradation occurs in the accuracy rate and that the model size is compressed so that it can be better deployed on edge devices, an edge device detection method is proposed in this paper based on ShuffleNetv2-YOLOv5-Lite-E for tea with one bud and two leaves. The first step was to build a normal YOLOv5 target detection model, the second step to remove the Focus layer, replace the original feature extraction network based on a ShuffleNetv2 algorithm, and then prune the YOLOv5 head on the neck layer to achieve the purpose of model size reduction. The third step was to build an environment on the edge device Raspberry Pi, so as to optimize the detection procedure, thus finally achieving an accurate and fast real-time detection of the improved YOLOv5 neural network through a camera connected to the edge device.

## 2. Materials and Methods

### 2.1. Dataset Production

2.1.1. Image Acquisition

The object of this study was Yunnan large-leaf tea trees. The images of tea trees used were taken from the base of Laobanzhang, Menghai County, Xishuangbanna Prefecture, Yunnan Province, China from 15 August to 27 September 2022, and a dataset was collected at the locations shown in Figure 1. Photographs were taken with a DSLR camera model SONY DSC-RX1RM2 while the photographer was standing 50–80 cm away from the tea trees. The distance between the camera and the tea with one bud and two leaves was 20–30 cm, the camera was 120 cm from the ground, and the vertical distance from the camera connected with the subsequent edge equipment to the ground was also 120 cm.

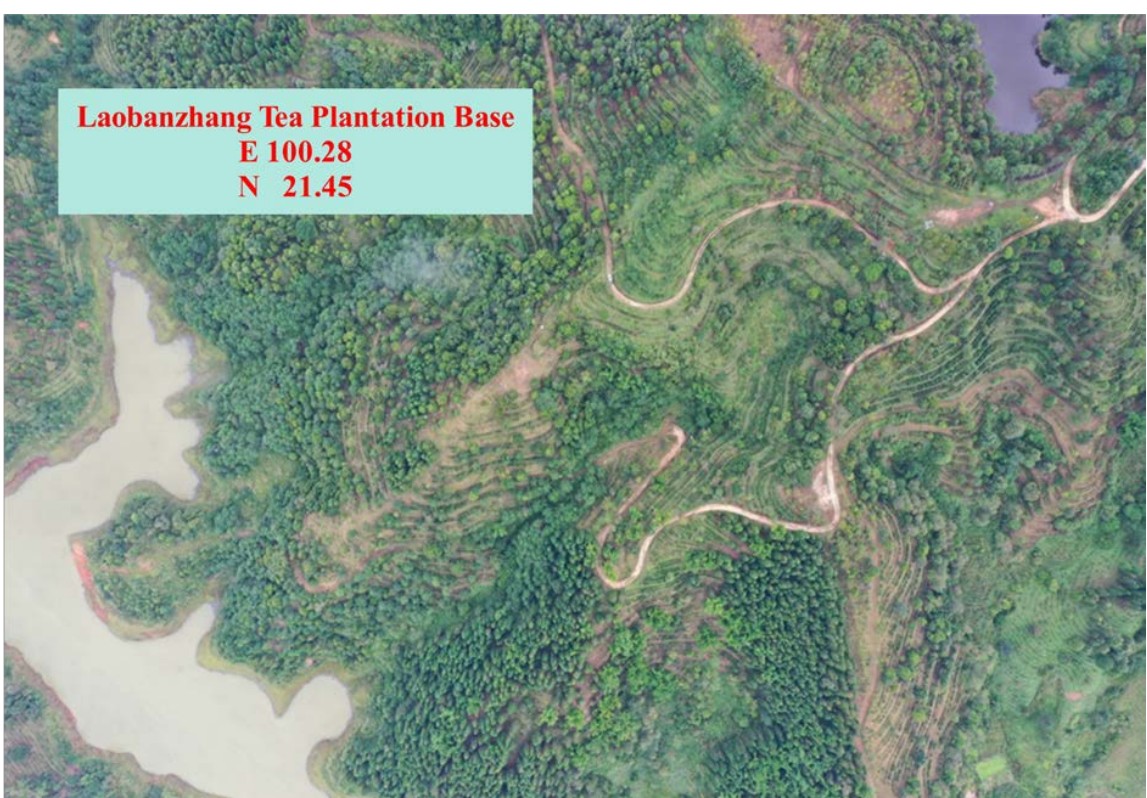

**Figure 1.** Data acquisition location.

The time of image acquisition was from 9:00 a.m. to 5:00 p.m., when the light intensity was slightly weaker in the morning and evening hours, which increased in the midday hours, resulting in the dataset containing different light intensities while ensuring the adaptability of the dataset with different light intensities after a model was generated for detection. In order to ensure the model generated via the neural network was accurately recognized in complex environments, the pictures of tea with one bud and two leaves

were collected by taking pictures of the same tea leaves once every 120 degrees. These were taken as the data base and for the important data for generating the model. In this study, data of different slopes and shaded surfaces of tea hills was collected to ensure the adaptability of this dataset to detect tea leaves with different slopes of tea hills after generating the model. The resolution of the images collected was 7952 pixels (horizontal direction) × 5340 pixels (vertical direction), the original images were saved in the jpg format, and a total of 2000 images was taken.

### 2.1.2. VOC Dataset Production

In order to reduce the influence of the dataset on the neural network model, the images collected were made into a VOC dataset in this study, with all the neural network models in the study being trained and tested. The images in the dataset were first screened, 1000 of which were selected for the dataset production. After that, the dataset was labeled using a labeling tool and the tea with one bud and two leaves was placed in the center of the label box during labeling. Eighty percent of the dataset was used as the training set (800 images) and 20% as the test set (200 images). Due to the complex and changeable environment and inconsistent light intensity of tea gardens, the dataset in this study contains images of tea with one bud and two leaves with different distances, different light intensities, and different backgrounds, which enhance the adaptability of the model to the complex environment of tea gardens to some extent. In this study, the recognition model generated by complex datasets is very beneficial for robots to work practically in tea gardens.

### 2.2. YOLOv5 Algorithm and Lightweighting Improvement

### 2.2.1. YOLOv5 Neural Network

The network structure of YOLOv5 is sketched in Figure 2. The whole YOLOv5 can be divided into three parts, namely Backbone, FPN, and YOLO Head.

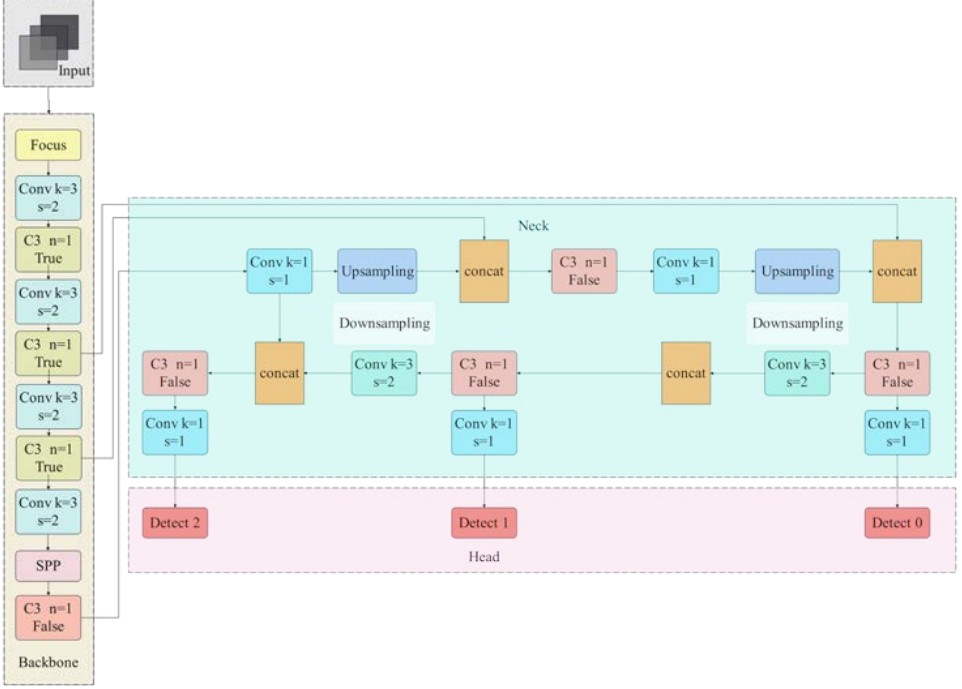

**Figure 2.** Basic structure diagram of YOLOv5.

Backbone is the backbone feature extraction network of YOLOv5, which is generally called CSPDarknet according to its structure and its previous name YOLO backbone. The input images are first extracted inside CSPDarknet for feature extraction, and the extracted features can be called feature layers, which means a set of features of the input

images [8]. In the backbone part, three feature layers are obtained for the next step of network construction, which are called effective feature layers.

FPN is called an enhanced feature extraction network of YOLOv5. The three effective feature layers obtained in the backbone part are feature-fused in this part, whose purpose is to combine feature information with different scales. In the FPN part, the effective feature layers already obtained will continue being used for feature extraction. The Panet structure is still used in YOLOv5, through which the features are both up-sampled and down-sampled again for feature fusion [9].

In the feature utilization section, multiple feature layers are extracted for target detection via YOLOv5, and a total of three feature layers is extracted.

The three feature layers are located at different positions of the trunk part CSPdarknet, namely the middle layer, the lower middle layer, and the bottom layer respectively. When the input is (640, 640, 3), the three feature layers have the shape of FEAT1 = (80, 80, 256), FEAT2 = (40, 40, 512), and FEAT3 = (20, 20, 1024).

After obtaining the three effective feature layers, the FPN layers are constructed by using them. A feature pyramid can be used to fuse the features of different-shaped feature layers for feature fusion, which is beneficial for extracting better features.

YOLO Head is the classifier and regressor of YOLOv5, through which three enhanced effective feature layers can be obtained through CSPDarknet and FPN. Each feature layer involves width, height, and number of channels, at which point the feature map is viewed as a collection of feature points one after another, with several channel features each. The yolo head actually works to determine whether there are objects corresponding to the feature points.

Therefore, the whole YOLOv5 network works as a feature–extraction–feature enhancement prediction of the object cases corresponding to the feature points.

The data enhancement method used in YOLOv5 is the Mosaic data enhancement method, which is an optimized extension of the CutMix data enhancement method [10,11], so unlike the CutMix data enhancement method where two images are stitched together, four images are stitched together with the Mosaic data enhancement method. A Mosaic data enhancement is accomplished by reading four images each time, and then flipping, scaling, as well as changing their color gamut, etc., and then arranging them in four directions. The use of Mosaic data enhancement greatly enriches the detection dataset, especially the random scaling, through which many small targets are added, making the network more robust. Since an RTX 3070 graphics card is used in this study—this can be used to directly calculate the data of the four images with very little space occupied by the mini-batches, while better results can be achieved with the use of one GPU—the Mosaic data enhancement method is more suitable for this study.

The loss of YOLOv5 consists of three components.

First of all, there is the Reg part. Each real box corresponds to a prior box. After obtaining the prior box corresponding to each box, the prediction box is taken out corresponding to the prior box, and the CIOU loss calculated using the real box as well as the prediction box, which is taken as the loss composition of the Reg part.

Second, there is the Obj part. Each real box corresponds to the a priori box. All real boxes corresponding to the a priori box are positive samples, while the remaining priori boxes are negative samples. According to the positive and negative samples as well as the prediction results of whether the feature points contain objects, the cross-entropy loss is calculated, which is taken as the loss composition of the Obj part.

Third, there is the Cls part. Each real box corresponds to the a priori box. After obtaining the a priori box corresponding to each box, the kind prediction result of the a priori box is taken out and the cross-entropy loss calculated according to the kind of real box and the kind prediction result of the a priori box, which is taken as the loss composition of the Cls part [12].

The $Loss_{CIOU}$ formula used for YOLOv5 is as follows:

$$Loss_{CIOU} = 1 - IOU + \frac{p^2\left(b, \ b^{gt}\right)}{c^2} + av \tag{1}$$

where $p^2\left(b, b^{gt}\right)$ represents the Euclidean distance between the centroids of the prediction frame and the true frame, $c$ represents the diagonal distance of the smallest closed region that can contain both the prediction frame and the true frame. The formulas for $\alpha$ and $v$ are as follows.

$$a = \frac{v}{1 - IOU + v}, \tag{2}$$

$$v = \frac{4}{\pi^2}\left(arctan\frac{w^{gt}}{h^{gt}} - arctan\frac{w}{h}\right)^2 \tag{3}$$

In the target recognition task, the objective optimization function is multi-peaked, and there are several local optimal solutions in addition to the global optimal solution. During training, the gradient descent algorithm may fall into its local minimum, and then the learning rate can suddenly increase to "jump out" of the local minimum and find a path to the global minimum. The value of the cosine function changes as x changes, the gradient descent first decelerates and then accelerates, which finally decelerates again. Through this method, we can avoid the loss value from entering the local optimal solution instead of the global optimal solution. For this reason, the cosine annealing decay method is used in this experiment to further reduce the loss value, where the cosine annealing decay method principle is as follows:

$$\eta_t = \eta_{min}^i + \frac{1}{2}\left(\eta_{max}^i - \eta_{min}^i\right)\left(1 + \cos\left(\frac{T_{cur}}{T_i}\pi\right)\right), \tag{4}$$

where $\eta_t$ is the current learning rate. $i$ is the value of the first few indexes run. $\eta_{max}^i$ is the maximum value of the learning rate. $\eta_{min}^i$ is the minimum value of the learning rate. $T_{cur}$ is the number of cycles in the current execution. $T_i$ is the total number of cycles in the current-run environment.

When the accuracy oscillation or loss value ceases to decay, the model is adjusted to achieve a lower loss value by adjusting the learning rate decay gradient through the cosine annealing decay method.

### 2.2.2. YOLOv5 Network Improvements

The main difference between the improved YOLOv5 network model and the original YOLOv5 network model is that the Focus layer is removed to avoid multiple-slice operations, a YOLOv4 feature extraction network is built using ShuffleNetv2 series [13], where the 1024 conv of backbone and 5 × 5 pooling are removed meanwhile the YOLOv5 head is pruned on the neck layer (Figure 3).

### Extraction of Focus

One Focus can replace several convolutional layers, but if the spatial information is focused on the channel space, the regression accuracy also reduces by 1 pixel, and most of the detection regression accuracy will not be close to 1, which is why Focus is placed on the 1st layer of the input.

At the same time, it is simple to understand that Focus speeds up to compress the network layer. But with the use of convolution instead of Focus, better performance was obtained, and the previous limitations or side effects did not appear, so this improved model was chosen to remove the Focus layer to avoid the use of multiple-slice operations. The frequent-slice operations are very serious for the cache occupation of chips, especially those without GPU or NPU acceleration, which increases the burden of computational processing. At the same time, it is more difficult to transform the Focus layer when the chips are deployed.

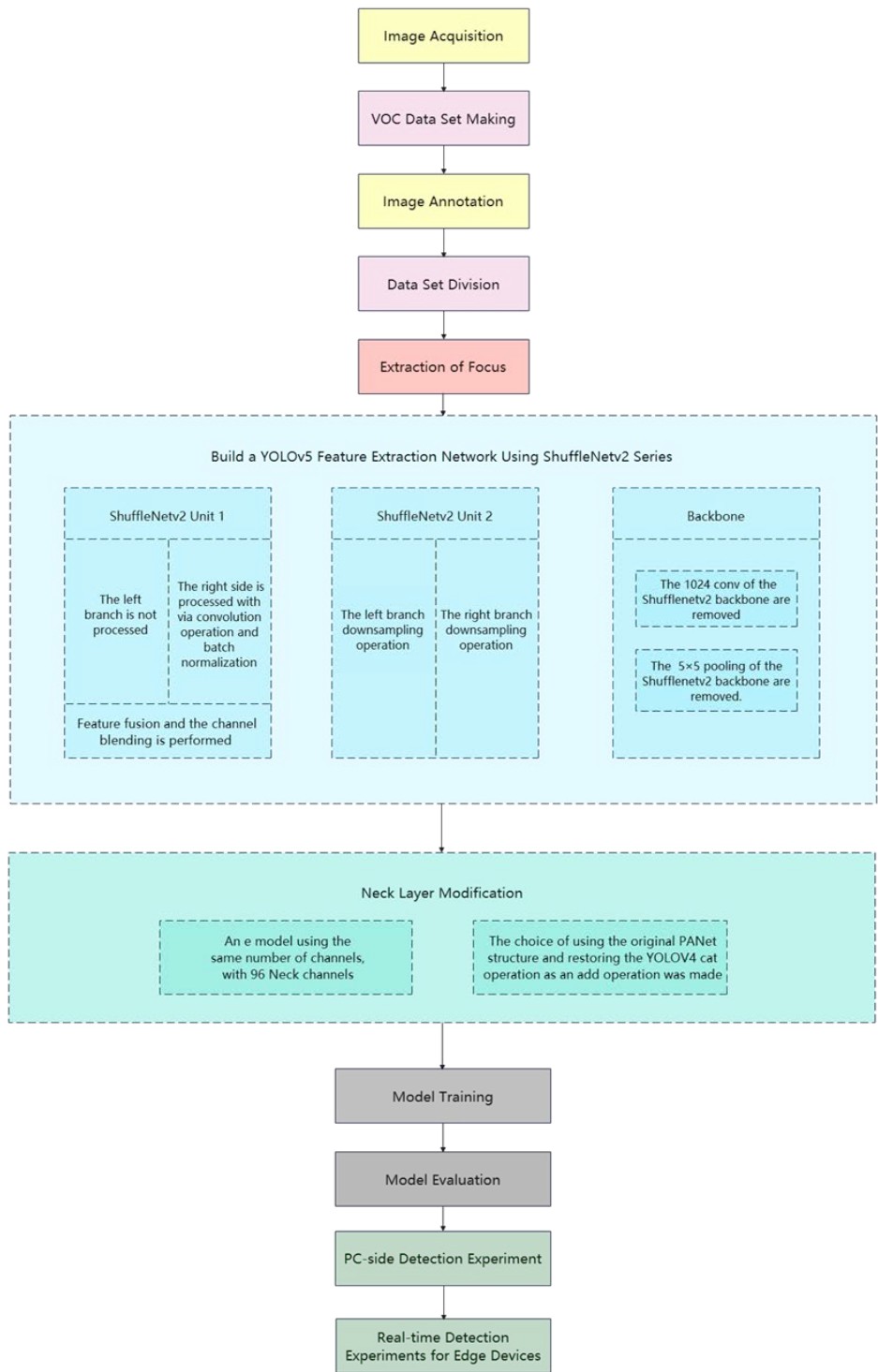

**Figure 3.** YOLOv5 network improvement step diagram.

Building a YOLOv5 Feature Extraction Network Using ShuffleNetv2 Series

The initial YOLOv5 model is prone to losing small target feature information in the feature extraction stage, so the detection of small targets is reduced. Its network model is large in size, involving a large number of parameters, and has high hardware requirements, which makes it difficult to be deployed on devices. To address the above situation, the backbone network in this study was replaced with a lightweight ShuffleNetv2 based on the initial YOLOv5 network to achieve the purpose of reducing parameters and achieving a lightweight network [14]. The feature communication among the feature maps of different

groups after group convolution is ensured by evenly breaking up the information of different channels of the output features obtained via group convolution, enhancing the feature extraction effect without increasing the computational efforts.

The layer structure of ShuffleNetv2 is shown in the figure, which is also the main structure of lightweight YOLOv5, corresponding to SFB1_X and SFB2_X in the structure diagram respectively. The number of input feature channels is divided into two groups by Unit 1, the left branch is not processed, while the right side is processed via convolution operation and batch normalization. Then the channel division features of the left branch are fused with the convolution output of the right branch, and channel blending is performed to enhance information fusion of the two channel maps. Through Unit 2, the left and right branch are down-sampled, the feature map size is halved, the dimension is doubled, and the ShuffleNetv2 network structure is formed by the two units together to achieve lightweight feature extraction (Figure 4).

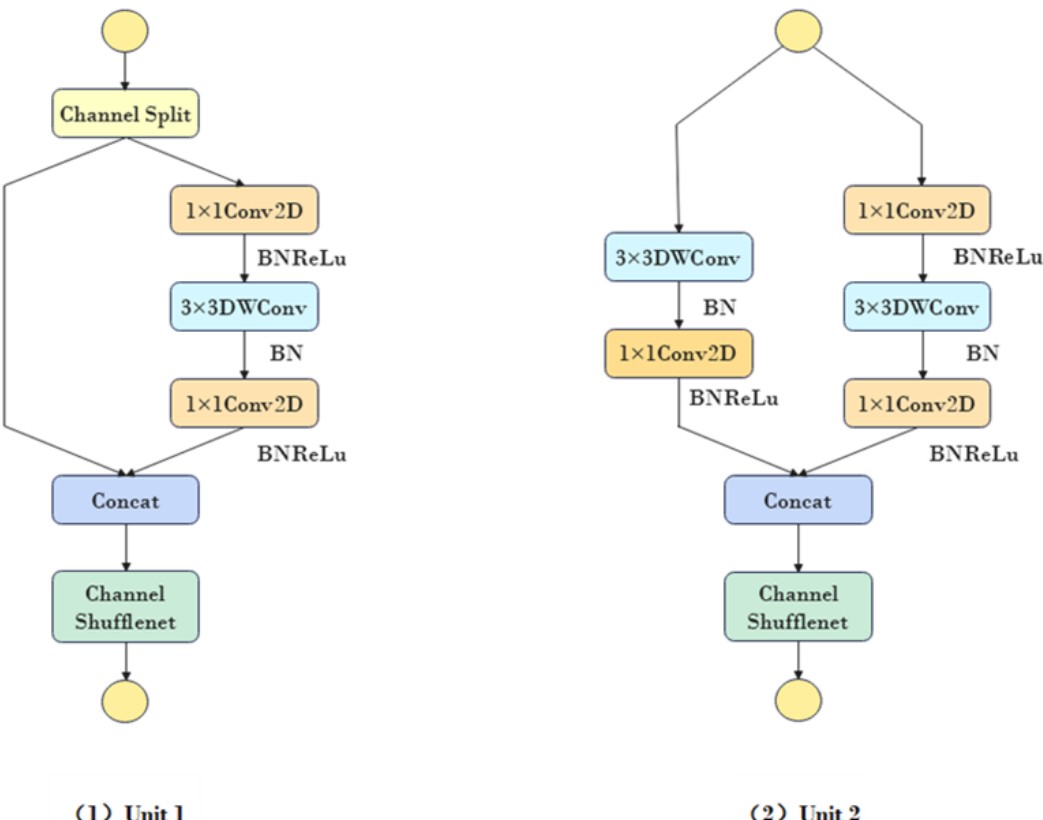

**Figure 4.** ShuffleNetv2 module diagram.

The channel separation operation has numerous advantages. The above structure is a case of not changing the number of input and output channels or the size of feature maps, while the pooling operation is performed using Unit 2 instead, and after such a structure, the number of image channels is expanded to twice that of the original. In addition, the 1024 conv and 5 × 5 pooling of the Shufflenetv2 backbone are removed from the Backbone.

Neck Layer Modification

In the field of target detection, some layers are usually chosen to be inserted in the Backbone and output layers, so as to better extract fused features, which are known as the Neck. A Neck is equivalent to the neck of a target detection network, through which the crucial links in feature fusion are extracted [15]. The structure of FPN + PAN is used in this improved YOLOv5 to prune the channels of the YOLOv5 head. The pruning rules refer to the design guidelines of ShuffleNetv2 while improving the structure of FPN + PAN through the previous version of YOLOv4, as follows.

An e model using the same number of channels, with 96 Neck channels, was chosen to optimize the memory access and usage to the maximum extent.

To further optimize memory usage, the choice of using the original PANet structure and restoring the YOLOv5 cat operation as an add operation was made (Figure 5).

Its activation function is as follows:

$$f(x) = x sigmoid(x) \tag{5}$$

SiLU is an improved version of Sigmoid and ReLU [16]. SiLU has no upper or lower bounds, which is smooth and non-monotonic. SiLU outperforms ReLU for deep models and can be seen as a smooth ReLU activation function.

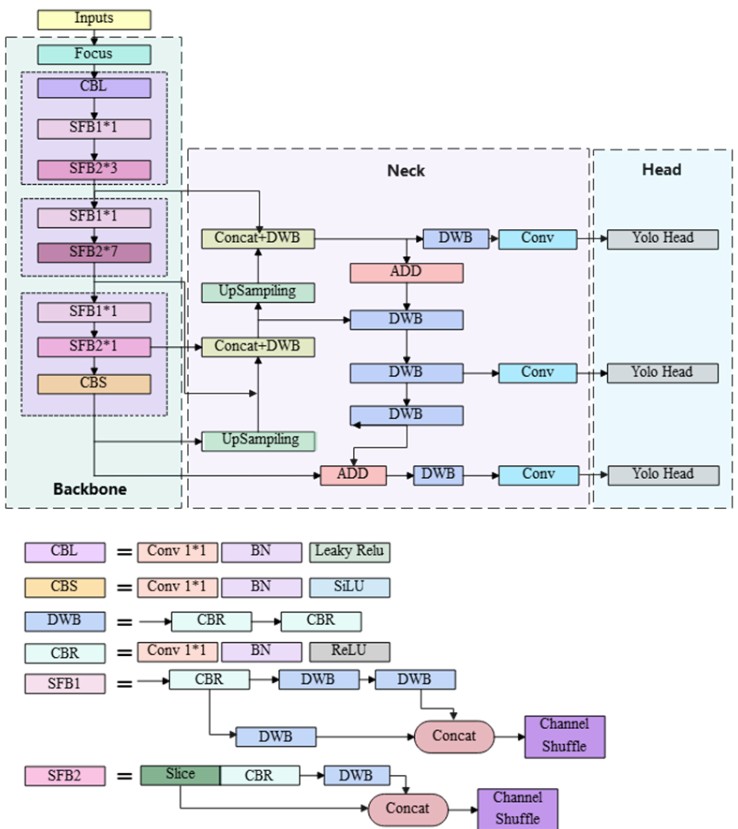

**Figure 5.** YOLOv5 network improvement diagram.

*2.3. Model Evaluation*

In this paper, the precision-recall curve (*P-R* curve) [17], AP (detection accuracy), mAP (average value of AP value for all categories), and detection speed are used as evaluation indexes. The AP value is an area enclosed by the *P-R* curve and the coordinate axis. mAP is the average value of AP in all categories. The recall (*R*) and precision (*P*) are calculated according to the following equations.

$$R = \frac{T_P}{T_P + F_N}, \tag{6}$$

$$P = \frac{T_P}{T_P + F_P} \tag{7}$$

$T_P$ is a positive class judged to be positive; $F_P$ is a negative class judged to be positive; and $F_N$ is a positive class judged to be negative.

AP and mAP are used to measure how well the model performs in each category and all categories respectively. Since this study is a single-category target detection recognition, the values of AP and mAP are the same.

## 3. Results and Analysis

### 3.1. Model Evaluation

Experiments were conducted using a TensorFlow-GPU2.4.1 deep learning framework, running deep learning on a host processor of the AMD Raider R7-5800H processor with 16GB RAM, 1TB hard drive, 16GB NAIDIA RTX 3070GPU, NVIDIA 461.37 driver, CUDA version 11.0, and a CUDNN neural network. The acceleration library version is 7.6.4. The algorithm was developed through Python 3.7 and Pycharm 2017. The edge device for the real-time detection step is a Raspberry Pi 4b, 4GB RAM, 16GB SD card storage, detection is performed using a Rasbian OS system, and the video real-time detection algorithm was developed through the Python package it carries.

In this example, a feature extraction network is fixed during the freeze training phase [18], to which only minor adjustments are made, so memory consumption is low; however, the backbone network of the model changes during the thaw training phase, and all of the network parameters change as well, so memory consumption increases significantly. This two-stage training scheme can be used to effectively reduce the amount of data to be processed during training, thus reducing the processor requirements on the model. The freeze training is set as 300 iterations, the batch size is set as 16, the momentum factor is set as 0.95, the decay factor is set as 0.005, and the initial learning rate is 0.01. The post-thaw training is set as 700 iterations, the batch size is set as 16, and the Mosaic data enhancement as well as the cosine annealing algorithm are used.

The size of the loss value is an important criterion for the assessment. Theoretically, the smaller the loss value, the better is the model training result. As is shown in Figure 6, the loss value decreases with the increase of training times. The loss value of the improved YOLOv5 network converge to below 1 after the 343rd iteration, which gradually becomes smooth after 550 iterations. A weight model is generated in each round of iteration, and a total of 1000 models is generated in this study, of which the optimal model is taken for subsequent detection and recognition.

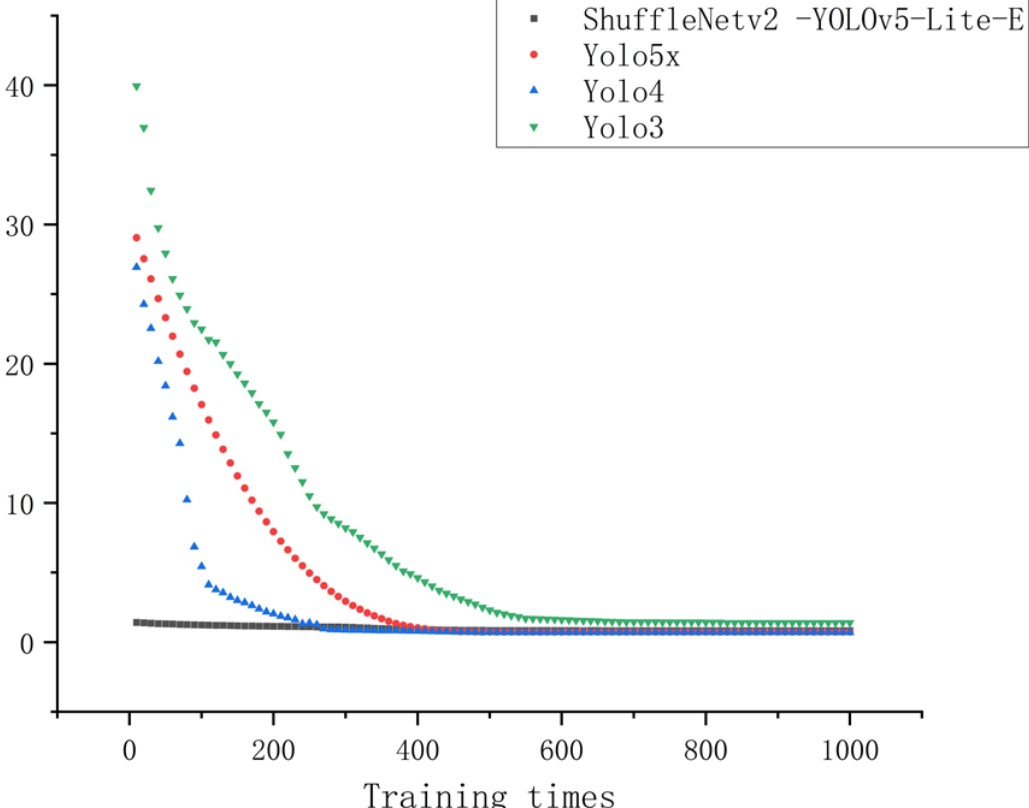

**Figure 6.** Comparison of loss changes.

The improved YOLOv5 network shows that the training set loss value (Train_loss) and the test set loss value (Val_loss) decrease as the number of training sessions increases, which indicates that the network training results are very good with no fitting occurring [19]. The stabilized Val_loss value is 0.1 higher than that of the YOLOv5 network model, 0.14 higher than that of the YOLOv4 network model, and 0.56 lower than that of the YOLOv3 network model (Figure 6).

### 3.2. Tea Identification Experiment

3.2.1. PC (Personal Computer)-Side Detection Experiment

In this study, the effectiveness of the ShuffleNetv2-YOLOv5-Lite-E network was validated by detecting and identifying tea with two leaves and one bud under different light intensities, with both single and multiple targets. Here, YOLOv5, YOLOv4, and YOLOv3 network models were trained and tested using the same dataset [20], while the platform configuration for training was consistent to ensure the authority of the results. Finally, the ShuffleNetv2-YOLOv5-Lite-E, YOLOv5, YOLOv4, and YOLOv3 deep learning models were used to compare the same dataset, and the comparison results are shown in Figure 7.

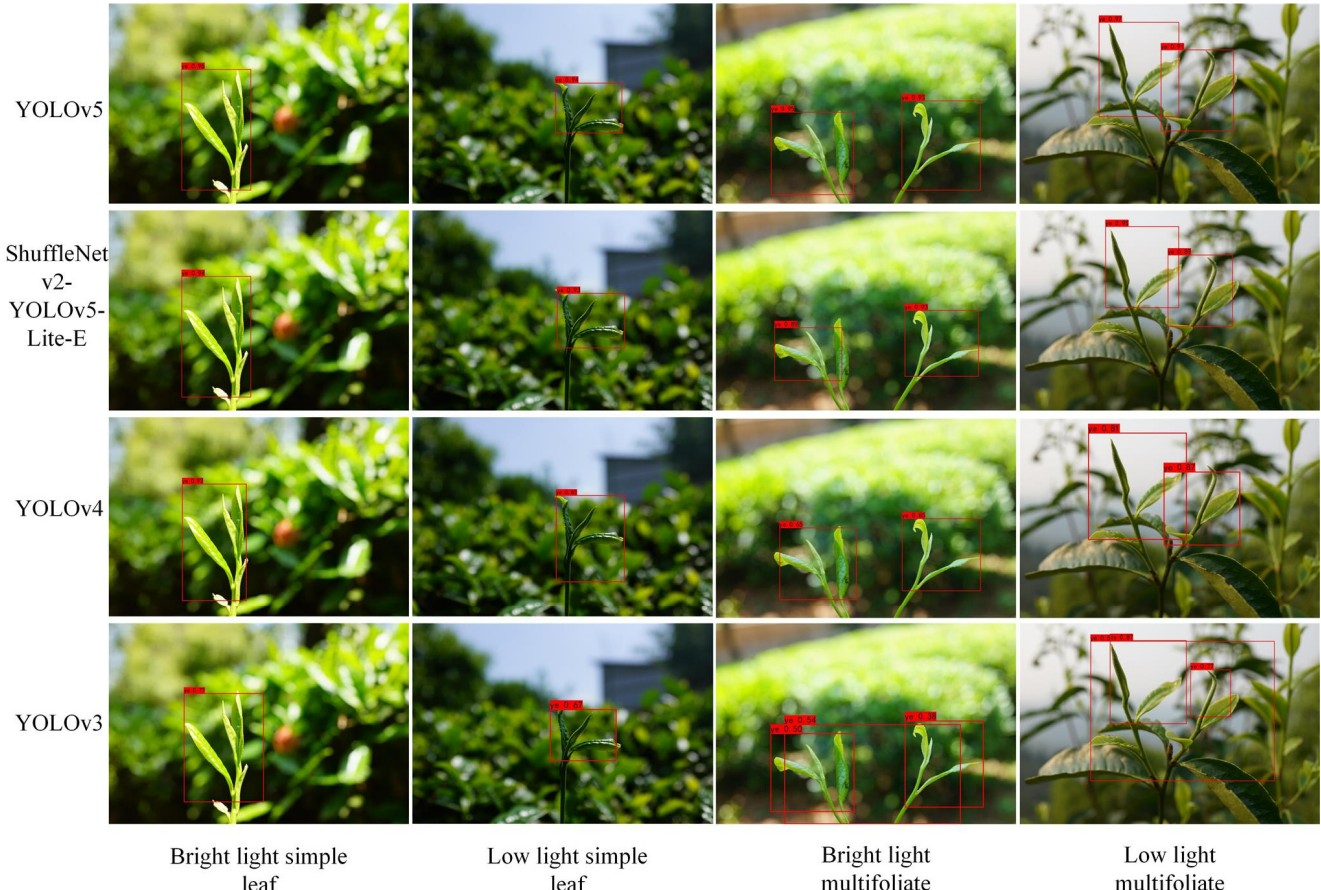

**Figure 7.** PC comparison results.

In the case of identifying tea with one bud and two leaves under strong illumination, all models tested could be used to complete the detection and identification of single and multiple targets, among which YOLOv5 and ShuffleNetv2-YOLOv5-Lite-E have the highest confidence in the detection results; the difference between them is less than 2%, while YOLOv3 has the lowest confidence in the detection results. In the case of multiple-target detection, YOLOv3 has the lowest confidence in the detection results, and when performing a multi-target detection [21,22], a serious omission is caused by YOLOv3, which also indicates that it has a low confidence in recognition (Figure 7).

All models tested could be used to achieve single-target and multi-target detection for the recognition of tea with one bud and two leaves in weak light, but their confidence decreases. The confidence of YOLOv5 and ShuffleNetv2- YOLOv5-Lite-E is still the highest, the difference between being less than 2% in misclassification. The experimental results show that the mAP value of ShuffleNetv2-YOLOv5-Lite-E model is 92.4%, for YOLOv5 93.7%, for YOLOv4 87.5%, and for YOLOv3 67.2% in low light.

Figure 8 presents a comparison of different mAP values of the four models. Because the detection experiments were conducted in a single category, the mAP values are also AP values. Table 1 shows a comparison of detection results and sizes of different training network models. From the table, we can see that the mAP value of the light-weighted and improved YOLOv5 is 1.32% lower than that of the original YOLOv5, but it is still 2.11% and 14.76% higher than that of YOLOv4 and YOLOv3, respectively, while the model size of ShuffleNetv2-YOLOv5-Lite-E is 27% that of YOLOv5, 75.0% that of YOLOv4, and 77.2% that of YOLOv3 (Table 1). The mAP values of ShuffleNetv2-YOLOv5-Lite-E model, YOLOv5, YOLOv4, and YOLOv3 are 97.43%, 98.75%, 93.56%, and 70.43% respectively.

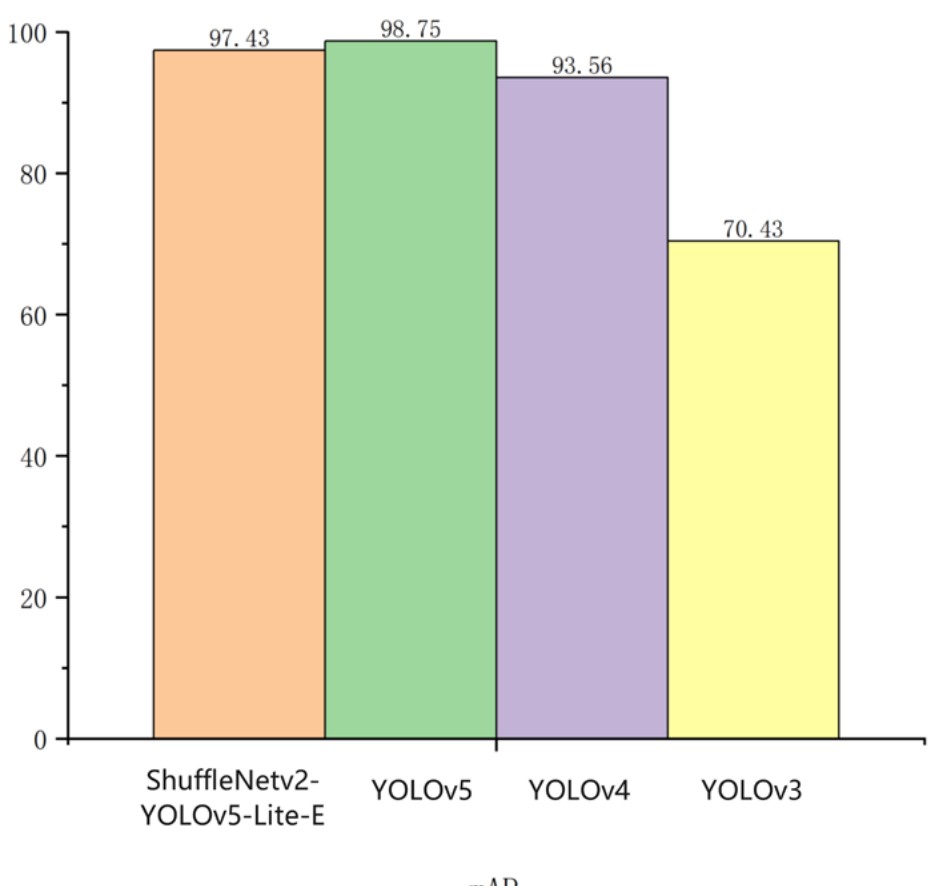

**Figure 8.** mAP comparison diagram.

**Table 1.** Model size and recognition rate.

| Model Name | Model Size/M | mAP |
|---|---|---|
| ShuffleNetv2-YOLOv5-Lite-E | 90.1 | 97.43 |
| YOLOv5 | 334 | 98.75 |
| YOLOv4 | 244 | 93.56 |
| YOLOv3 | 237 | 70.43 |

3.2.2. Real-Time Detection Experiments for Edge Devices

A Raspberry Pi 4b model with 4GB running memory and 16GB storage space was adopted for the plug-in SD card of the edge device, in which the official Rasbian OS system that comes with Raspberry Pi was adopted and the models ShuffleNetv2-YOLOv5-Lite-E, YOLOv5, YOLOv4, as well as YOLOv3 were imported. The system comes with the Python library to write a calling camera program and then a real-time recognition program that can fit the calling of the four models, respectively, so as to complete the setting of the edge device. The camera is a 10-megapixel camera with a USB interface, which is plugged into the USB interface of Raspberry Pi as the video transmission tool for real-time recognition. The Raspberry Pi with the camera is fixed on a crawler car, and the camera is 1.2 m from the ground to maintain a uniform height with data collection, which is consistent with the dense tea area of the inspection site.

The detection site this time is a tea garden of Yunnan Agricultural University. When conducting the experiment, four different model programs were called in the same scene respectively, each of which intercepted 100 images while running. Table 2 displays their average recognition rate as well as recognition speed.

**Table 2.** Identification comparison chart.

| Model Name | Number of Photos Taken/Piece | Average Confidence/% | Detection Speed/Frame |
| --- | --- | --- | --- |
| ShuffleNetv2-YOLOv5-Lite-E | 100 | 94.52 | 8.6 |
| YOLOv5 | 100 | 96.27 | 2.7 |
| YOLOv4 | 100 | 93.56 | 3.2 |
| YOLOv3 | 100 | 70.43 | 3.4 |

In this study, the recognition model generated by complex datasets is very beneficial for robots to work practically in tea gardens. When edge equipment works in a tea garden, the captured images are also pictures with different lighting, distances, and backgrounds. The research results show that the recognition rate is low, generally between 85% and 90%, and that when the camera is far away from the two leaves of a bud, which are partially blocked, the recognition rate is extremely high, generally above 95%, while when the camera is close to the two leaves of a bud, the illumination intensity is sufficient. Therefore, it is necessary to control the distance between robots and the tea trees as much as possible in the subsequent application of the model. In view of the dark weather, portable light sources can be used for the work, which can improve the recognition rate of robots to a certain extent.

When performing recognition, the difference between the confidence of ShuffleNetv2-YOLOv5-Lite-E and YOLOv5 is only 1.75%, which is slightly higher than the average confidence of YOLOv4 and is larger than that of YOLOv3. However, in terms of detection speed, ShuffleNetv2-YOLOv5-Lite-E is 3.2 times faster than YOLOv5, 2.7 times faster than YOLOv4, and 2.5 times faster than YOLOv3. Through this experiment, it can be seen that ShuffleNetv2-YOLOv5-Lite-E has the fastest recognition speed and the highest corresponding frame rate while ensuring the confidence level, with the frame rate roughly being maintained at 8.6 frames/s, which fully satisfies the need of real-time detection (Table 2).

## 4. Conclusions

A target detection model is proposed in this study based on ShuffleNetv2-YOLOv5-Lite-E, which removes the Focus layer and replaces the original feature extraction network via the ShuffleNetv2 algorithm, followed by channel pruning of the YOLOv5 head at the neck layer, thus achieving the purpose of model size reduction. The improved ShuffleNetv2-YOLOv5-Lite-E network is more robust, and the network model is more lightweight,

through which the detection efficiency of the device can be improved while ensuring that the recognition rate is not reduced.

According to the actual operating environment of the experimental tea plantations, image datasets including different light intensities and angles were produced and tested using ShuffleNetv2-YOLOv5-Lite-E, YOLOv5, YOLOv4, and YOLOv3 models respectively. The results showed that the detection speed and model size of ShuffleNetv2-YOLOv5-Lite-E network model were better than those of other networks under conditions with a guaranteed detection accuracy. YOLOv5-Lite-E network model outperforms other networks in terms of detection speed and model size with a guaranteed detection accuracy.

The ShuffleNetv2-YOLOv5-Lite-E network model was evaluated for its comprehensive performance, through which not only the recognition detection of tea with one bud and two leaves can be completed in complex situations with large changes in light brightness, but its recognition detection can also be accomplished at different angles. The ShuffleNetv2-YOLOv5-Lite-E network model can be used to accurately carry out the recognition of tea with one bud and two leaves in the case of both single and multiple recognition. On an embedded device, an average accuracy of 94.52% was achieved through the model for the recognition of tea with one bud and two leaves with a detection speed of 8.6 frames/s, making it ideal for transplanting to edge devices for real-time recognition detection.

**Author Contributions:** Conceptualization, writing—original draft preparation, S.Z., H.Y. and C.Y.; methodology, W.Y.; software, X.C.; validation, X.W. and Y.S.; formal analysis, X.D. and W.H.; investigation, L.L. and J.H.; data curation, Y.Z.; conceptualization, writing—review and editing, funding acquisition, B.W. and X.L. All authors have read and agreed to the published version of the manuscript.

**Funding:** This research was funded by National Key Research and Development Program (2022YFD1601800) and Major Special Science and Technology Project of Yunnan Province (202002AE090010).

**Institutional Review Board Statement:** Not applicable.

**Informed Consent Statement:** Not applicable.

**Data Availability Statement:** Publicly available datasets were analyzed in this study. This data can be found here: [https://github.com/anqi99/ShuffleNetv2-YOLOv5-Lite-E-2-.git (accessed on 7 February 2023)].

**Conflicts of Interest:** The authors declare no conflict of interest.

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
