# Peer review of "Edge Device Detection of Tea Leaves with One Bud and Two Leaves Based on ShuffleNetv2-YOLOv5-Lite-E"

_agronomy, doi:10.3390/agronomy13020577_

Round 1

Reviewer 1 Report

This paper presents a DL model named "Shuf-

fleNetv2-YOLOv5-Lite-E" for the detection of tea with one bud and two leaves. The results show that the improved model size is 27% smaller than the original YOLOv5 model, with a mAP value of 97.43% on a PC and 94.52% on an edge device. My main comments are presented below.

Major:

The work must present a more in-depth related work discussion using YoloV5 and ShuffleNet. Some relevant works are presented in [1] and [2]. Include all the references you will find in the state-of-the-art.

Given that [1] and [2] state better results than yolov5s. I highly recommend you include them in assessing the proposal of [1] and [2].

There are new versions of yolo. For example, yolov8s outperformed yolov5s in coco dataset. Include a paragraph mentioning the advantage or motivation to improve the architecture of yoloV5 rather than newer versions of yolo. 

Considering how the images were acquired (every 120 degrees), it is possible that the camera was focused only on the object of interest. The authors could explain if the network is detecting the tea leaves or if the network is detecting the non-blurry section of the image. The Fig. 7 looks like the tea leaves are sharp and the rest of the image is blurry. I highly suggest including gradCam to visualize the attention of yolo.

Data Availability Statement indicates that the data is publicly available in a github repository, but it was not. I suggest using a better site to store your public data. 

Minor:

Include the amount of waste on line 49.

The paragraph in lines 54-64 is one sentence. I suggest dividing it into several sentences to improve the readability.

Include the amount of data used for the training/validation/testing in the works mentioned in lines 75-82. It would be useful for the reader to understand if the performance of the detectors is related to the amount of data used.

Include the latitude and longitude in the axis of Figure 1.

Check the typos in the manuscript. For example, "Foucs" in Fig. 2. Read carefully and find all the typos.

Reduce the length of the sentences to make them easier to understand. For example, lines 54-64 are a single sentence. 

There is an inconsistency in the GPU used, the line 197 mentioned RTX 3070 but line 321 mentioned "NAIDIA RTX 3060". Correct the typo.

Change Figure 6 for a 2D plot. It would be easier to see the differences between the curves.

Include the number of the Figure you are referring to. For example, lines 270 and 361. Check all the text.

[1]. Wang, L., Zhao, Y., Xiong, Z., Wang, S., Li, Y., & Lan, Y. (2022). Fast and precise detection of litchi fruits for yield estimation based on the improved YOLOv5 model. Frontiers in Plant Science13.

[2]. Li, Y., Li, A., Li, X., & Liang, D. (2022, August). Detection and Identification of Peach Leaf Diseases based on YOLO v5 Improved Model. In Proceedings of the 5th International Conference on Control and Computer Vision (pp. 79-84).

Author Response

Thanks very much for your time to review this manuscript. I really appreciateyou’re your comments and suggestions. We have considered these comments carefully and triedour best to address every one of them.

1.The work must present a more in-depth related work discussion using YoloV5 and ShuffleNet. Some relevant works are presented in [1] and [2]. Include all the references you will find in the state-of-the-art.Given that [1] and [2] state better results than yolov5s. I highly recommend you include them in assessing the proposal of [1] and [2].

Modification instructions: In response to your comments, we have compared our improved method with the above two and have taken these two studies as our references.Have been changed in accordance with the advice given.

2. There are new versions of yolo. For example, yolov8s outperformed yolov5s in coco dataset. Include a paragraph mentioning the advantage or motivation to improve the architecture of yoloV5 rather than newer versions of yolo.

Modification instructions: Have been changed in accordance with the advice given.

3. Considering how the images were acquired (every 120 degrees), it is possible that the camera was focused only on the object of interest. The authors could explain if the network is detecting the tea leaves or if the network is detecting the non-blurry section of the image. The Fig. 7 looks like the tea leaves are sharp and the rest of the image is blurry. I highly suggest including gradCam to visualize the attention of yolo.

Modification instructions: All parts of the images are recognized through YOLOv5, including the blurred parts. The reason why we compare the pictures with blurred backgrounds is that YOLOv3 has a poor recognition efficiency and a high recognition error probability for pictures without blurred backgrounds. In order to facilitate the comparison, we use blurred pictures. Thank you very much for suggesting the use of Grad-CAM for visual display. However, due to the epidemic situation and the Chinese New Year, we have not returned to the institute to work at present, where the related computers are temporarily unavailable. I am very sorry about this, and I have added an identification of pictures with an unverified background before our model and the photos of the tracked robots used in our field test in the attachment to prove the application of our improved model in practice. If Grad-CAM is really needed for visual display, I will apply for returning to the institute for testing as soon as possible. I am really sorry.

4. Data Availability Statement indicates that the data is publicly available in a github repository, but it was not. I suggest using a better site to store your public data. 

Modification instructions: The final model and dataset have been uploaded again.

5. Include the amount of waste on line 49.

Modification instructions: Have been changed in accordance with the advice given.

6. The paragraph in lines 54-64 is one sentence. I suggest dividing it into several sentences to improve the readability.

Modification instructions: Have been changed in accordance with the advice given.

7. Include the amount of data used for the training/validation/testing in the works mentioned in lines 75-82. It would be useful for the reader to understand if the performance of the detectors is related to the amount of data used.

Modification instructions: Have been changed in accordance with the advice given.

8. Include the latitude and longitude in the axis of Figure 1.Check the typos in the manuscript. For example, "Foucs" in Fig. 2. Read carefully and find all the typos.

Modification instructions: Have been changed in accordance with the advice given.

9. Reduce the length of the sentences to make them easier to understand. For example, lines 54-64 are a single sentence. 

Modification instructions: Have been changed in accordance with the advice given.

10. There is an inconsistency in the GPU used, the line 197 mentioned RTX 3070 but line 321 mentioned "NAIDIA RTX 3060". Correct the typo.

Modification instructions: Have been changed in accordance with the advice given.

11. Change Figure 6 for a 2D plot. It would be easier to see the differences between the curves.

Modification instructions: Have been changed in accordance with the advice given.

12. Include the number of the Figure you are referring to. For example, lines 270 and 361. Check all the text.

Modification instructions: Have been changed in accordance with the advice given.

Reviewer 2 Report

Dear authors,

This paper presents a very interesting application of using a modified YOLOv5 to detect tea leaves with one bud and two leaves. One of the main contributions of your work is the lightweight modification of the network to make it more usable on edge devices. The potential impact of this work can be high for future robot tea leaf picking. The overall topic and work you have done should be enough for publication. However, the quality of the paper presentation should be further improved before publishing. 

The key problem is that the key points of the presentation should be focused on what's new in your study. I would pick two. 1. The application scenario, which is the tea bud detection. 2. the modification of the network. You did fine on point 2 but put a very small effort into describing 1. In the meantime, a large part of the text is used to describe YOLOv5 which is not a contribution of your work.

Specifically: 

1. The description on YOLOv5 should be largely distilled. 

2. Put more figures on this specific tea dataset, and show the reader a better picture of what data are in the dataset.

3. Show more figures on the detection results on this dataset. Such as showing the typical failure cases and success cases under various conditions (light, time, cam pose, etc.)

4. Why do you use this type of chart in figure 8? The x-axis seems meaningless.

Author Response

Thanks very much for your time to review this manuscript. I really appreciateyou’re your comments and suggestions. We have considered these comments carefully and triedour best to address every one of them.

In response to your questions, we have added detailed data of multi-model comparisons in different application scenarios, increased comparisons with other latest improvements on YOLOv5, added the photos of tracked robots when conducting field tests in the annex, replaced the relevant pictures, and reduced the introduction of the original YOLOv5. Thank you very much for your advice.

1.The description on YOLOv5 should be largely distilled. 

Modification instructions: Have been changed in accordance with the advice given. The basic framework of YOLOv5 is introduced in the uncut parts, which can make it easier for readers to understand our improvement ideas and schemes. If still needed, we can make further cuts.

2. Put more figures on this specific tea dataset, and show the reader a better picture of what data are in the dataset.

Modification instructions: Have been changed in accordance with the advice given. In response to your questions, we have added a more detailed introduction of datasets and the reasons for adopting complex datasets in the paper.

3. Show more figures on the detection results on this dataset. Such as showing the typical failure cases and success cases under various conditions (light, time, cam pose, etc.)

Modification instructions: Have been changed in accordance with the advice given.

4. Why do you use this type of chart in figure 8? The x-axis seems meaningless.

Modification instructions: Have been changed in accordance with the advice given. In response to your questions, we changed the pictures in the paper for readers' better comparisons.

Round 2

Reviewer 1 Report

The authors have replied properly to all my comments.

I suggest that you check the grammar and typos that are still in the current version of the manuscript. For example, "Pie" in line 61.